

# High resolution acoustic recordings of wild free-ranging short-beaked common dolphins for etho-acoustical and repertoire studies

Loïc Lehnhoff[1,2,3,*], Hervé Glotin[2,3,*], Yves Le Gall[4], Eric Menut[4], Hélène Peltier[5], Jérôme Spitz[5], Olivier Van Canneyt[5], and Bastien Mérigot[1,3,*]

[1]MARBEC, Univ Montpellier, CNRS, IFREMER, IRD, Sète, France
[2]Université de Toulon, Aix Marseille Univ, CNRS, LIS DYNI, Toulon, France
[3]Centre international. d'IA en Acoustique Naturelle (CIAN), France
[4]French Research Institute for Exploitation of the Sea (IFREMER), Centre Bretagne, Plouzané, France
[5]Observatoire Pelagis, La Rochelle Université, CNRS, La Rochelle, France
[*]These authors contributed equally to this work.

**Correspondence:** Loïc Lehnhoff (loic.lehnhoff@gmail.com)

**Abstract.** Dolphins are highly vocal cetaceans with a complex acoustic repertoire. These marine mammals rely heavily on sound for critical activities: echolocation clicks for navigation and prey detection, whistles for social communication, and pulsed sounds for less well-documented purposes. Understanding their acoustic behaviour is essential for insights into their ecology, social structure, and responses to anthropogenic noise. However, to date, there is a lack of open-access datasets of

acoustic recordings of wild free-ranging short-beaked common dolphins (*Delphinus delphis*), coupled with observations data. Here, we present a new dataset (DOI: 10.5281/zenodo.14637674, Lehnhoff (2025)) of high resolution acoustic recordings of (*D. delphis*) observed during various behavioural states, including foraging, travelling, socializing, milling, and attraction to the boat. The dataset was collected in the northern Bay of Biscay, France, from summers of 2020 to 2022 during surveys conducted as part of the DOLPHINFREE project. The dataset contains acoustic recordings of wild free-ranging short-beaked

common dolphins (*Delphinus delphis*) observed during various behavioural states, including foraging, travelling, socializing, milling, and attraction to the boat. Audio recordings were performed during opportunistic encounters using two devices: a single high-quality hydrophone (sampling rate: 512 kHz, bit-depth: 32 bits) and a compact array of four hydrophones (256 kHz to 512 kHz, 16 to 24 bits) for localization purposes. The dataset comprises over 400 minutes of unedited audio recordings of *D. delphis* accompanied by visual observations. In total, we identified about 68,000 echolocation clicks, 4,600 whistle contours,

and more that 350 pulsed sounds. This comprehensive resource is valuable for detailed studies of the acoustic repertoire of common dolphins and their two-dimensional movements.

Keywords: etho-acoustic, bio-acoustic, cetaceans, echolocation click, whistle, buzz, burst pulse, *Delphinus delphis*



# 1 Introduction

The study of the sounds emitted by cetaceans represents a field of research with numerous applications, including the investigation of social interactions, behavioural patterns, localisation (including trajectory reconstruction) and the identification of these animals (e.g. Caldwell and Caldwell, 1968; Overstrom, 1983; Poupard et al., 2019; Halkias et al., 2013). Cetaceans have different and distinctive vocal repertoires between species, populations (e.g. Ansmann et al., 2007) or even between individuals within a single group (e.g. Fearey et al., 2019; Cones et al., 2023). This data descriptor makes available a bio-acoustic dataset of audio recordings of wild, free-ranging short-beaked common dolphins (*Delphinus delphis*) collected at sea. Comparable datasets are available for other delphinid species (e.g. Di Nardo et al., 2023). However, to date, there is a lack of open-access datasets of acoustic recordings of *D. delphis*. The present study proposes a high-sampling, high-resolution audio dataset accompanied by visual observations, providing a comprehensive contextual framework for each recording.

Short-beaked common dolphins emit sounds that are similar to other delphinid species. Acoustic signals emitted by delphinid species, including short-beaked common dolphins, can be divided into three main categories: echolocation clicks, whistles and rapid sequences of clicks (pulsed sounds, often referred to as buzzes and/or burst-pulses Jones et al. (2020)). The echolocation clicks of these animals are short broadband pulses, which enable dolphins to effectively navigate their surrounding environment and to recognise the nature of objects (Norris et al., 1961; Tyack, 1986; Au, 1993). Whistles are narrow-band frequency-modulated signals that are thought to be used for inter-individual communication (Caldwell and Caldwell, 1968; Au and Hastings, 2008), individual identification (signature whistles) (Sayigh, 1992) and coordination of group movements (Lammers and Au, 2003; Branstetter et al., 2012). Finally, these dolphins also emit rapid sequences of clicks, that form an other form of communication: pulsed sounds. Buzzes have been linked to hunting behaviours (Wisniewska et al., 2014; Ridgway et al., 2015), while burst-pulses are frequently observed during social interactions (Overstrom, 1983; Lammers et al., 2006). The distinction between these signals is based on their inter-click interval (ICI) (Martin et al., 2019). However, the function of these pulsed signals remains subject to debate (Ridgway et al., 2015).

The dataset that we provide is extracted from audio recordings collected during surveys at sea performed in the frame of the DOLPHINFREE project off the coast of Penmarc'h, Brittany, France in 2020, 2021 and 2022. The aim was to visually and acoustically assess the behavioural responses of wild, free-ranging common dolphins to a bio-inspired acoustic signal, emitted by the version 1 of the CETASAVER-DOLPHINFREE beacon (OCTech company), the purpose of which is to reduce the risks of fishery bycatch (Lehnhoff et al., 2022). Therefore, acoustic recordings were made with and without the activation of the beacon, but also in presence/absence of a fishing net, and depending on the behavioural state of the dolphins (i.e. foraging, travelling, socialising, milling, attraction to the boat), in order to assess the acoustic activities of the animals in different conditions. Only the acoustic recordings collected before the beacon was turned ON are made available, as (i) the DOLPHINFREE signals are protected by intellectual property rights (property of the University of Montpellier and IFREMER), and (ii) in order to deliver data that is not dependent on the activation of the device, enabling the study of the natural behaviour of the dolphins during the different states cited above.



This data descriptor therefore presents a selection of the acoustic data collected during the DOLPHINFREE surveys, together with the visual observations performed at the same times. Two types of devices were used to make acoustic recordings (see detailed specifications in the Materials and Methods section): (i) a single Ocean Sonics icListen high-quality hydrophone deployed from an observation boat, and (ii) a custom-made antenna (TETRA), which is a compact tetrahedral array made of 4 hydrophones, suspended under a buoy (see fig 1). The aim of the recordings made using the icListen hydrophone was to record the signals of dolphins from the observation boat to access their acoustic behaviour, while the aim of the recordings made with the TETRA antenna was more dedicated to studying the movements of the dolphins in two dimensions during the surveys. As these devices were not designed for the same purposes, they were often deployed during different sequences. In total, 275 and 162 minutes of active recording made with the icListen hydrophone and the TETRA antenna, respectively, are made available. For these recordings, visual observations confirmed the presence of dolphins around the hydrophones. These data provide a wide panorama of the signals that short-beaked common dolphins can emit: echolocation clicks, whistles, pulsed sounds and probable bi-phonations (Jones et al., 2020) can be found in the dataset.

This dataset provides complete audio recordings of encounters with wild short-beaked common dolphins at sea, in the Bay of Biscay, which should be of interest for the study of the acoustic repertoire of these animals, their whistles, their acoustic behaviours, their localisation using acoustic signals as well as tasks of whistle detection.

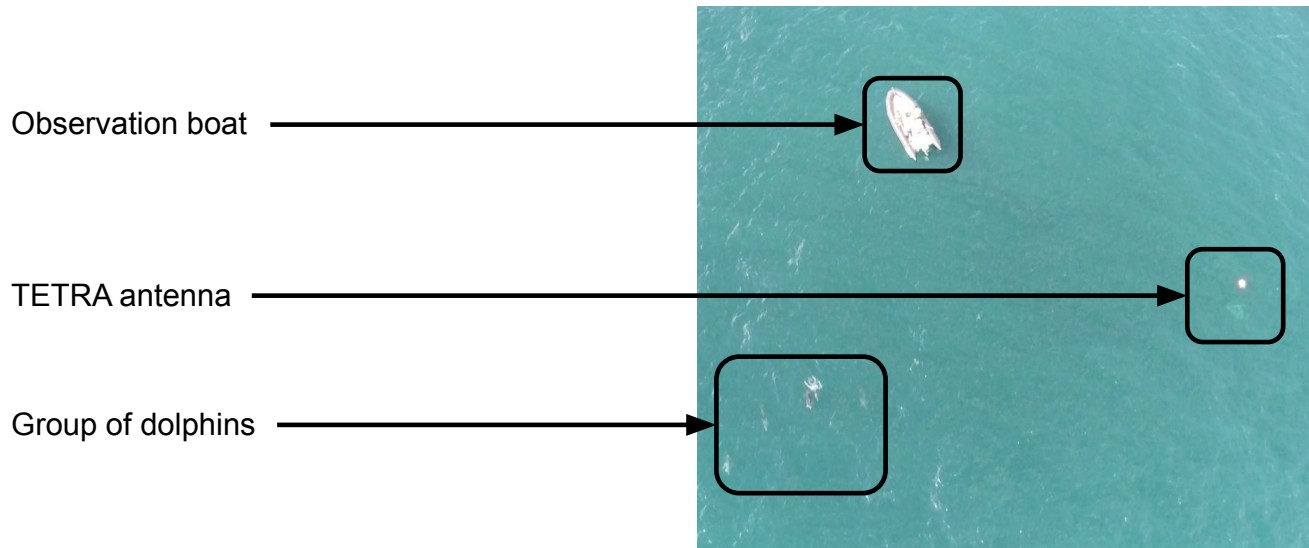

**Figure 1.** Typical example of the experimental layout during the DOLPHINFREE surveys. *Aerial image captured by a DJI Phantom drone piloted by B. Mérigot.*





## 2    Material and surveys

### 2.1    Study area

Non-systemic scientific surveys were carried out at sea to find dolphins in opportunistic encounters off the coast of Penmarc'h,
Brittany, France (see Figure 2). Wild dolphins are frequently sighted in this area, just a few nautical miles off the coast. In
addition, the occurrence of dolphin strandings in the area (Peltier et al., 2020), contributed to the selection of this site. Surveys
were performed on 11–17 July 2020, 9–18 July 2021 and 16–24 July 2022 when the weather conditions allowed visual sightings
of the dolphins and the identification of their behaviours from semi-rigid pneumatic boats carrying visual observers: with a
wind ≤ 10 knots and a swell ≤ 1 m.

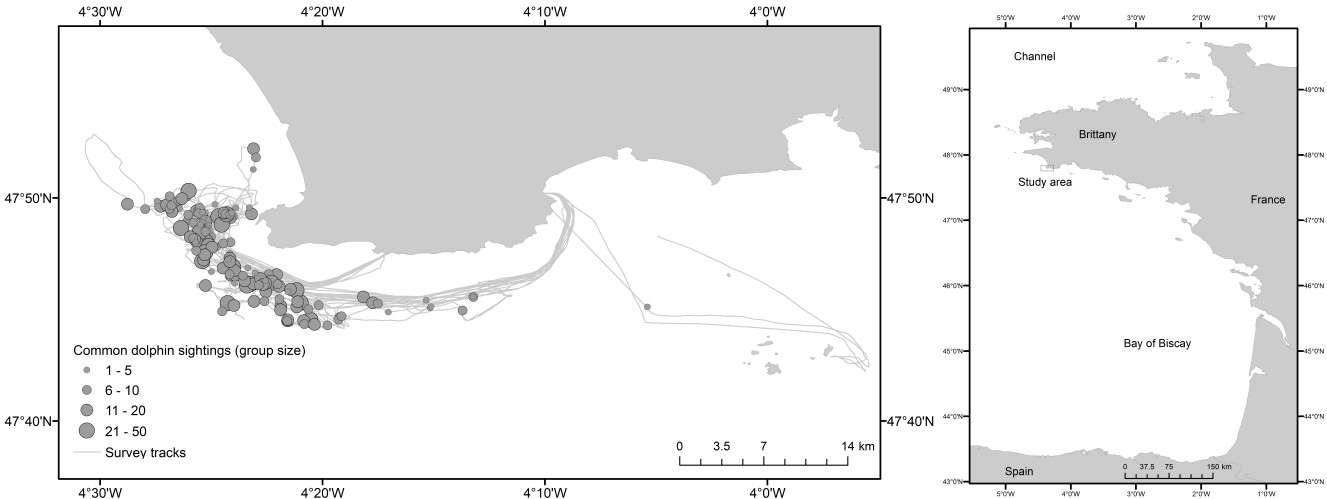

**Figure 2.** Map of dolphins encountered during the 2020, 2021 and 2022 DOLPHINFREE surveys.

### 2.2    Acoustic devices

#### 2.2.1    Main hydrophone

The Ocean Sonics icListen HF hydrophone (see appendix B1 for technical details) was used during surveys at sea in order to
record short-beaked common dolphins acoustic signals. This high-frequency hydrophone was used to record sounds on one
channel, with a sampling rate of 512 kHz and an audio bit-depth of 32-bits. When the observation boat was stationary (engine
off), this hydrophone was deployed from one side of the boat and positioned at -3 m underwater. These parameters allow for a
high quality sampling of the acoustic signals in the vicinity of the boat, without any computational processing.

### 2.2.2 Compact hydrophone array

A custom-made prototype of a compact array of 4 hydrophones was deployed during the surveys. The device (hereafter called TETRA, Figure 3) has a tetrahedral shape, with one hydrophone at each one of its apexes: one CR3 spherical hydrophone with linear frequency range up to 180 kHz, and three SQ26 cylindrical hydrophones with satisfactory responses up to 50 kHz (see

appendix B2 & B3 for technical details). The array is made of PVC tubes joined by 3D-printed parts, and connected to a QHB motherboard made by the SMIoT laboratory (Barchasz et al., 2020), University of Toulon, France (technical specifications available online here).

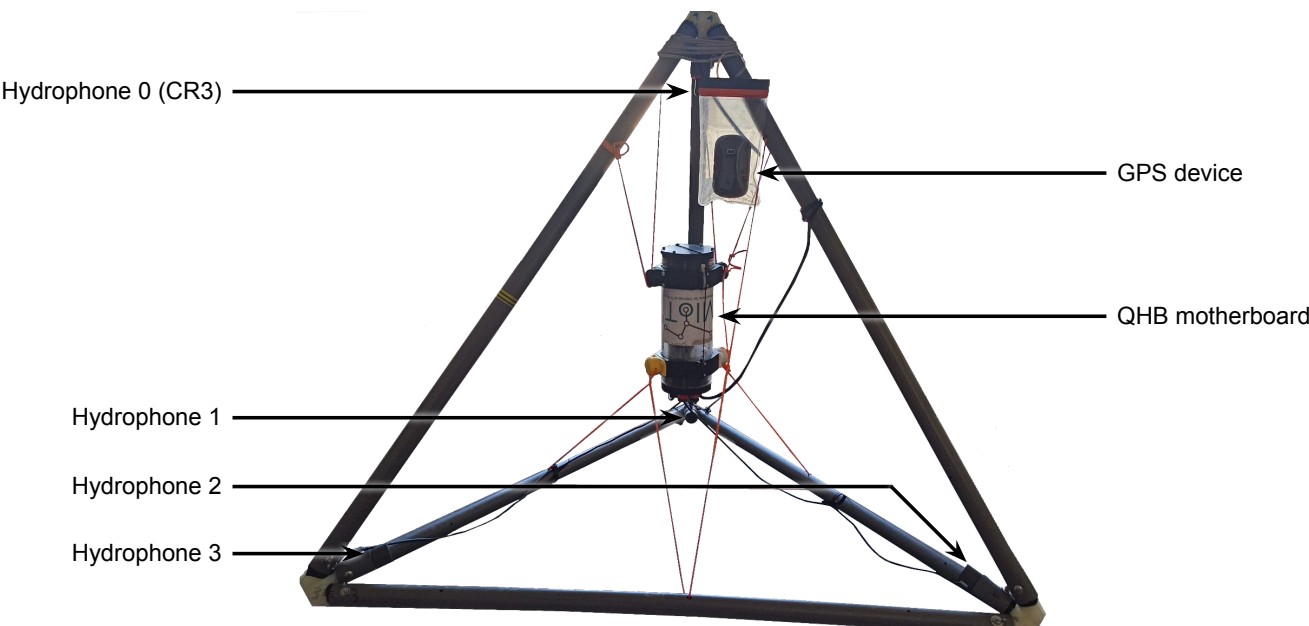

**Figure 3.** Annotated photo of the TETRA antenna used during the 2021 and 2022 surveys of the DOLPHINFREE project. TETRA's sides are ≈90 cm in length. *Photograph taken by L. Lehnhoff before the surveys.*

This array was used in 2021 and 2022 to record audio on 4 channels at 256 kHz to 512 kHz, with a bit-depth of 16 to 24 bits, depending on survey sessions. Its materials and its size (about 90 cm in length) make it a compact, portable and practical

device. Once in the water, it is left to drift freely under a buoy at -3 m underwater, and is monitored by observers onboard. Consequently, when both recording devices were deployed, audio recordings made with TETRA might differ from those of the icListen hydrophone, due to the distance between the two devices. It should also be noted that this version of TETRA was an initial prototype, which has since been improved in subsequent versions (see Glotin et al. (2024)).



## 2.3 Survey design

During the surveys, two to three people were present on board to navigate, to deploy the acoustic equipment, and to take notes of the visual observations made on the observed groups of dolphins. A group of dolphins was defined as any number of animals observed within five body lengths of another conspecific, moving and behaving in the same pattern (Shane, 1990; Stockin et al., 2008; Filby et al., 2013). The distances of each group of dolphins from the boat were initially estimated using rangefinder binoculars (Bushnell Fusion $10\times42$) that enabled visual observers to get accustomed to estimating distances at

sea. Once the observers were used to estimating distances, the binoculars were set aside as this allowed for quicker visual observations, which was a more pragmatic approach.

The DOLPHINFREE surveys required a group of dolphins to show a constant behavioural state (defined in Table 1) for at least 1 min of observation. As soon as this condition was met, the boat was stopped, the engine and sonar were switched off. Then, the recording devices were set: the icListen hydrophone at -3 m from one of the sides of the boat, and the TETRA

antenna at -3 m under a buoy left to drift from the other side of the boat. However, it should be noted that the TETRA antenna was only deployed when drift conditions enabled the observers on board to maintain visual contact with the buoy.

**Table 1.** Definitions of behavioural states of common dolphins recorded in 2020, 2021 and 2022 (according to (Van Canneyt et al., 2006; Berg Soto et al., 2013; Filby et al., 2013), from (Lehnhoff et al., 2022)).

| State | Definition |
| --- | --- |
| Foraging | Dolphins involved in any effort to pursue, capture and/or consume prey, as defined by observations of two or more of the following: fish chasing; erratic movements at the surface; multi-directional diving; coordinated deep diving; and rapid circle swimming. Preys and hunting birds often observed at the surface. |
| Travelling | Dolphins engaged in persistent, directional movement making noticeable headway along a specific compass bearing. Group spacing varied and individuals swam with short (<20 s), relatively constant dive intervals. |
| Socialising | Animals were involved in active surface behaviour (frequent surfacing and breaching) that included physical interactions among group members and sometimes aerial behaviour. |
| Milling | Dolphins showed little movement, tended to remain in the same place and either spent floating at the surface or surfaced asynchronously. |
| Attraction | Dolphins came towards the boat and swam at a few metres along it, following its direction. |

As stated in the Introduction section, only the recordings made before the emission of signals by the DOLPHINFREE acoustic beacon are available here. However, other signals were tested during part of the surveys: classical music pieces, whistles of orcas. The audio recordings of these sequences are fully available. In addition, during some surveys, a fishing net

was set underwater to simulate the conditions in which bycatch occur. The distribution of audio files according to the behaviour observed and the presence or absence of a fishing net is presented in Figure 4.



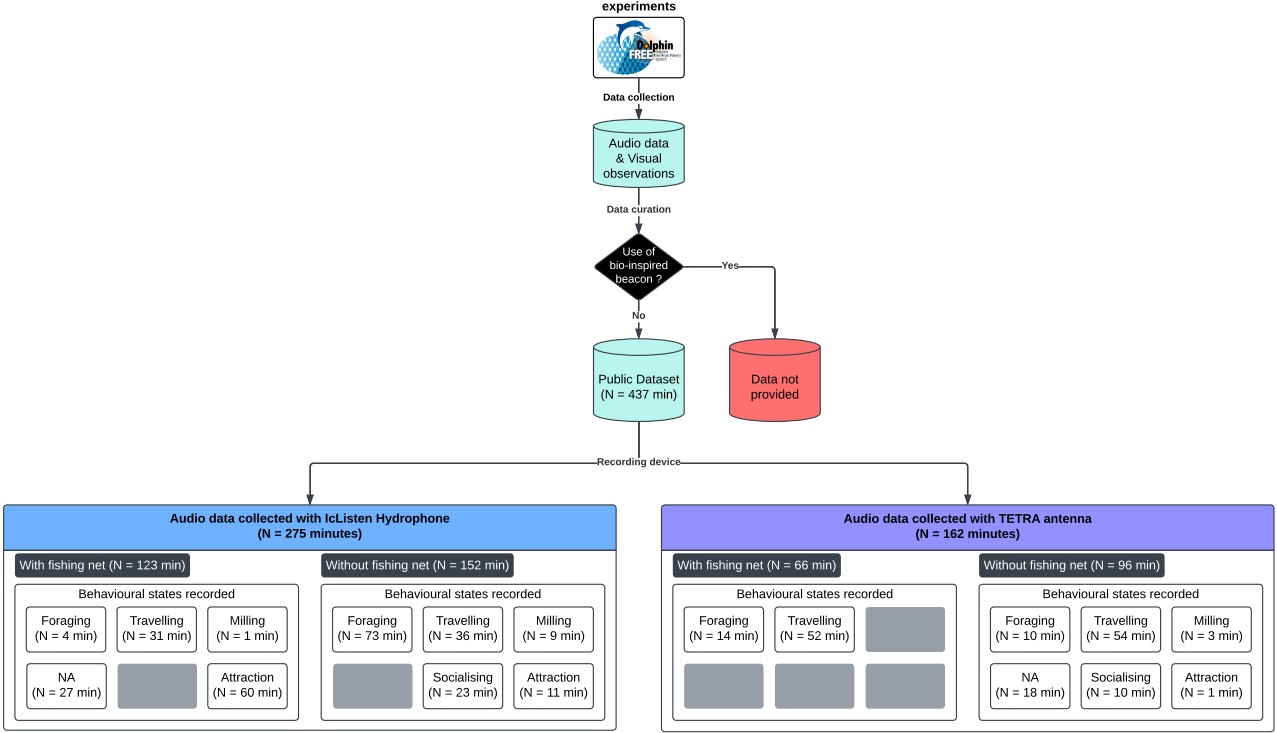

**Figure 4.** Diagram illustrating the distribution of data collected as part of the DOLPHINFREE surveys, with a focus on the data made available depending on the presence of a fishing net, and the observed behavioural state of the dolphins.

## 3 Data & methods

This section describes the files and data types that will be found in the dataset (DOI: 10.5281/zenodo.14637674, Lehnhoff (2025)) that is made available.

### 3.1 Audio files

These are raw files obtained from the recording of wild free-ranging short-beaked common dolphins during the DOLPHIN-FREE surveys. There are 275 unedited 1-minute files of audio recordings made using the icListen hydrophone. These files have a sampling rate of 512 kHz with 32-bits resolution on 1 channel.

As the tetrahedral hydrophone array (TETRA) is a custom-made device, we experimented with different configurations of its QHB motherboard, leading to audio recordings made with varying parameters. In total, 117 files are available, which cumulated, add up to 162 minutes of audio data. Files obtained from the TETRA antenna were recorded on 4 channels (cor-





responding to its 4 hydrophones) and have varying sampling rates (256 kHz or 512 kHz), bit-depth resolutions (16 to 24 bits) and durations (15 s to 120 s).

For the acoustic data recorded from both recording devices, we recommend the application of a high-pass filter for the study of echolocation clicks, in order to avoid background noises (such as waves and boat engines), as both devices were deployed only a few meters below the water surface.

## 3.2  Observation notes

Observation data was collected manually during the surveys, following a custom-made data table (see appendix A). Then, notes were standardised and associated with each audio file independently. These visual observations are available as .xlsx

files, distinct for each year and acoustic recording device. Each file contains the values described in Table 2).

**Table 2.** Signification of column names in XLSX files of visual observations.

| Column name | Signification | Value format |
|---|---|---|
| audio_file | Name of the wavefile. | [date]_[time].wav |
| datetime_utc | Start date and time of recording. | YYYY-MM-DD HH:MM:SS |
| observers | Names of observers taking notes. | firstname_lastname |
| ID_group | Identification number of a group of dolphin. | Integer |
| ID_sequence | Identification number of each survey sequence. | Integer |
| group_size | Number of individuals in group. | Integer |
| fishing_net | Presence/absence of a fishing net. | "present" or "absent" |
| fishing_net_type | When present, type of fishing net deployed. | "type_of_net" |
| behaviour | Observed behavioural state of the group of dolphins. | see Table 1 |
| behavioural_event | Special behavioural events. | jumps, spyhopping, ... |
| distance | Distance from the boat to the closest visible dolphin. | Integer |
| group_clustering | Aspect of the spacing between individuals. | "compact" or "scattered" |
| direction | General direction in which the group is heading. | "continuous" or "variable" |
| speed | Average speed of the animals (slow/fast threshold: ≈10 kn). | "slow", "variable" or "fast". |
| diving_time | Average diving time of the animals (normal/long threshold: 2 min). | "normal", "variable" or "long" |
| activation_sequence | Experimental treatment (relative to the emission of signals). | "before", "during", "after" or "control" |
| signal | Type of signal loaded in the emitter. | see `signal_codes.txt` file |
| sonar_noise | Indicates if a sonar was unintentionally recorded. | 0 (No), 1 (Yes) |
| special_observations | Any additional observation | Notes |





### 3.3 Whistle contour annotation

The shapes of whistles can be linked to specific behaviours and contexts (Lehnhoff et al., 2022) or directly to individuals (i.e. signature whistles (Caldwell and Caldwell, 1965)). However, the extraction of whistle contours represents a challenging task, for which a variety of automated methods have been developed (e.g. using modelling (Halkias and Ellis, 2006; Roch et al.,

2011), pitch-tracking (Baumgartner and Mussoline, 2011), or deep learning (Conant et al., 2022; Li et al., 2023) techniques).

We used a semi-automated method (Lehnhoff et al., in review.a) to annotate the contour of whistles from audio recording made using the IcListen hydrophone. The spectrograms used for the annotations were generated using the following parameters: 96,000 Hz sampling rate, 1024 samples (11 ms) frame size, 512 samples (5 ms) hop length, on a linear frequency scale. These annotations were then manually verified and corrected using PyAVA (Lehnhoff, 2022), a custom-made annotation tool for

whistle contours. These results are made available as .json in the **Whistle_annotations** in the dataset.

### 3.4 Other data

A **README.md** file is available in the parent folder of the dataset and describes the structure of the dataset, with usage notes and links to academic papers produced using its data.

The **Tetra\Hydro_coordinates** folder contains the .csv files giving the measured coordinates of the 4 hydrophones

of the tetrahedral antenna in 2021 and 2022. These coordinates are needed to determine the angle of arrival of the signals detected using the hydrophones.

### 4  Results & output data

Some analyses were already conducted on this dataset and are published/in review/submitted in 3 different scientific papers. These papers show an increase in the acoustic activity of common dolphins in response to the DOLPHINFREE bio-inspired

acoustic signal (Lehnhoff et al., 2022), characterize the features of whistles (Lehnhoff et al., in review.a) and identify signature whistles in the whistling repertoire of these animals (Lehnhoff et al., in review.b). However, it should be noted that the dataset made available here is slightly different than the one used in our previous analyses (due to the exclusion of audio recordings containing DOLPHINFREE signals).

From these works, the technical quality of the dataset is supported by manual and automatic detections of dolphins signals

in audio recordings. A standard Teager-Kaiser click detector coupled with a data projection to exclude false positive detections (Lehnhoff et al., 2022) was used to confirm the presence of echolocation clicks in most files. A semi-automatic detector of whistles was also applied to extract contour coordinates of whistles (Lehnhoff et al., in review.a) with a manual validation of the detections. In total, we confirm the presence of about 68,000 echolocation clicks, 4,600 whistles and 350 pulsed sounds in the provided audio data recorded with the icListen hydrophone (Lehnhoff et al., 2022, in review.a).

The distribution of dolphin signals across the recordings is quite heterogenous. Figure 5 showcases the different signals that can be found in the audio files made available. In addition, we show the magnitude spectrum of a sample of echolocation clicks recorded by the icListen hydrophone at 512 kHz, revealing the broadband nature of these clicks (see Fig 6).

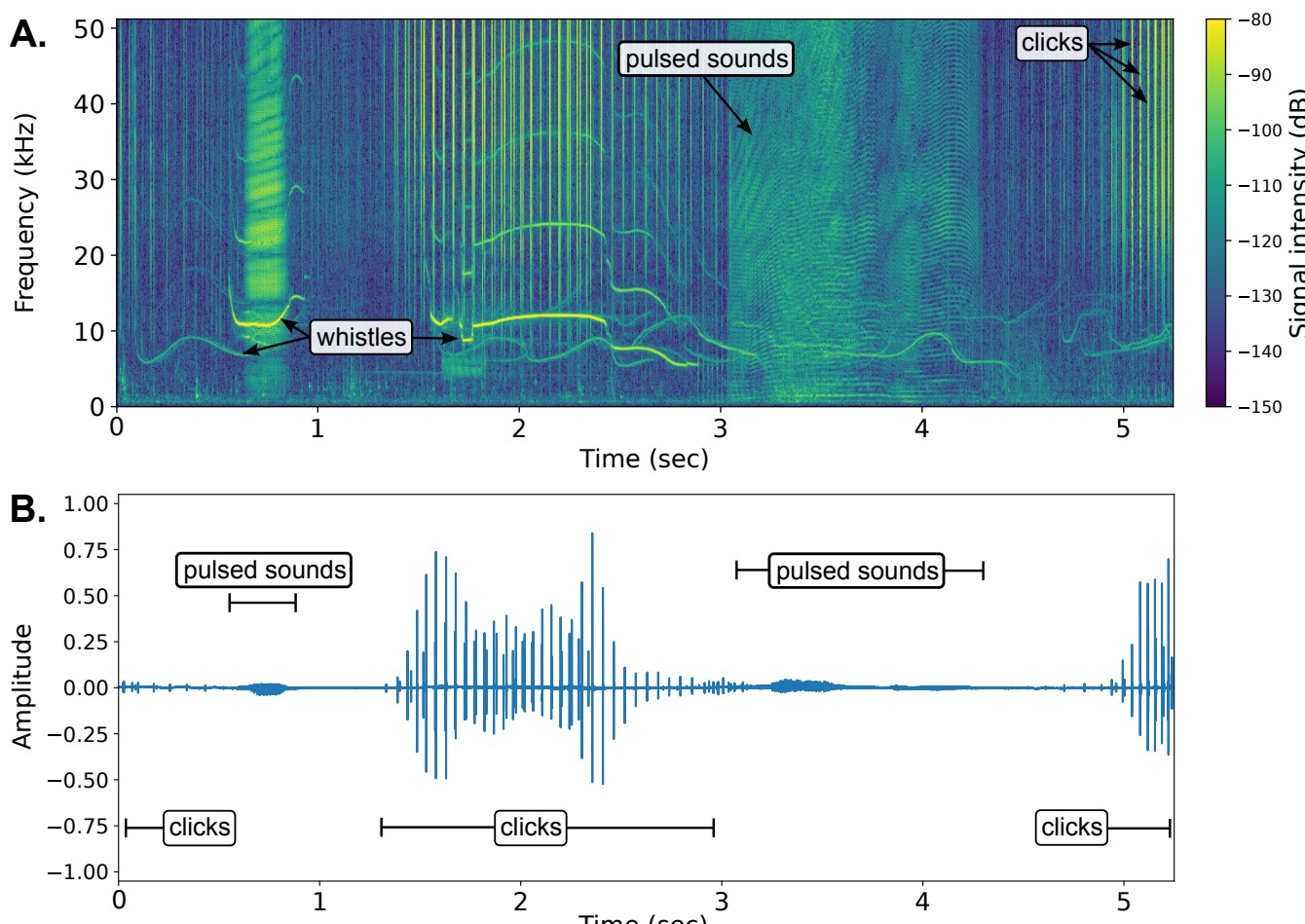

**Figure 5.** Spectrogram (**A**) and waveform (**B**) extracted from audio recording "SCW1807_20200713_064400.wav".

*(A) Arrows point to representative signals. B Segments show the timespan of click-like sounds. The annotations indicate only some of the signals emitted by short-beaked common dolphins that are visible on the spectrogram.*




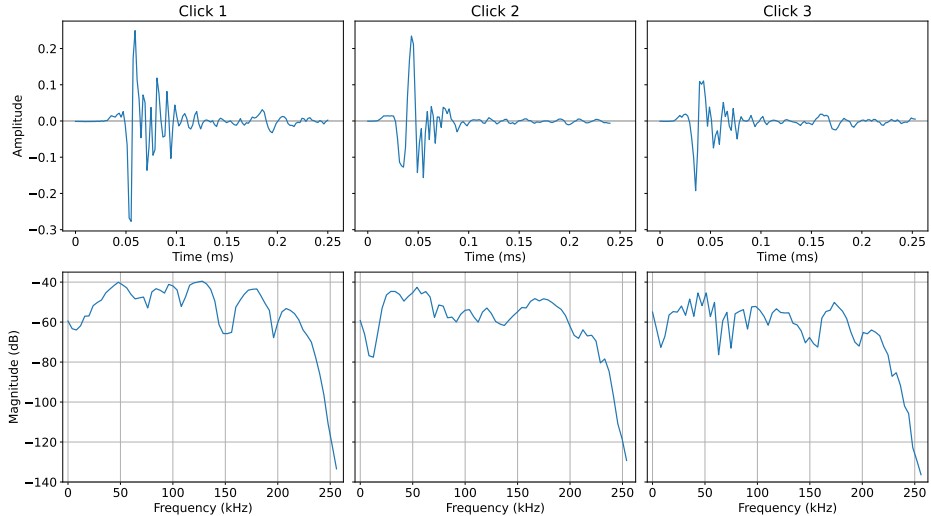

**Figure 6.** Waveforms and magnitude spectrums of 3 echolocation clicks selected randomly in audio file "SCW1807_20200713_064400.wav".

## 4.1 2D localisation

TETRA's recordings (4 channels, see Figure 7) can be used to determine the time difference of arrival (TDoA) of echolocation

clicks to the hydrophones of the antenna, also enabling the estimation of the angle of arrival (AoA) of those clicks. A validation sequence, to evaluate the errors made in the estimation of the AoAs, was conducted. DOLPHINFREE signals were emitted from the boat and recorded by the TETRA antenna at different angles around it.

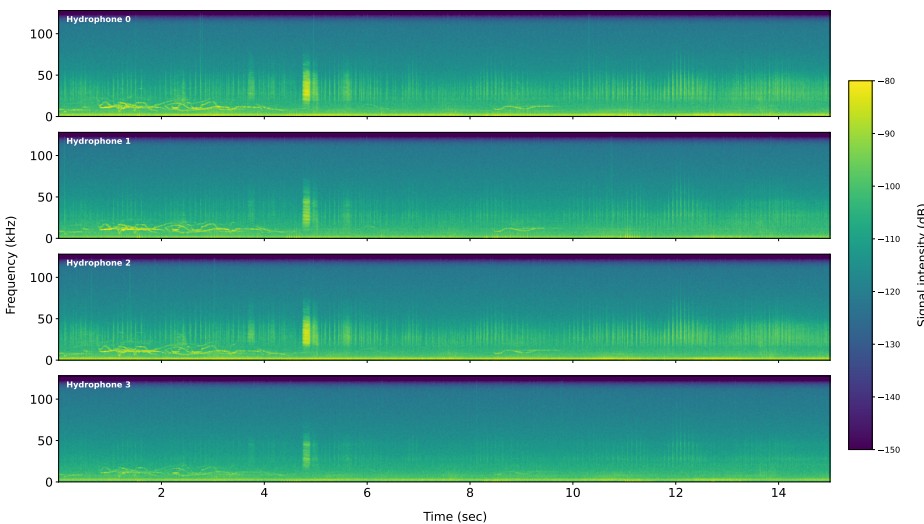

**Figure 7.** Spectrogram extracted from audio recording "20210709_135634UTC+2_V12.wav" collected with the TETRA antenna. Hydrophone 0 is a CR3, other hydrophones are SQ26.





The position of the antenna relative to the observation boat was determined by two GPS devices: one located on the observation boat and the other attached to the TETRA antenna buoy. Then, using the audio recordings, the DOLPHINFREE signals were used to estimate AoAs from TDoAs. A comparison of the values estimated by these two methods is shown in Fig. 8. It highlights that there are only a few deviations in the estimation of AoAs using TETRA, compared to GPS measurements (on average 0.15 radians (8.6°) for the azimuths and 0.08 radians (4.4°) for the elevation angles). However, since calibrations were made by recording the DOLPHINFREE signal, the corresponding audio recordings are not publicly available.

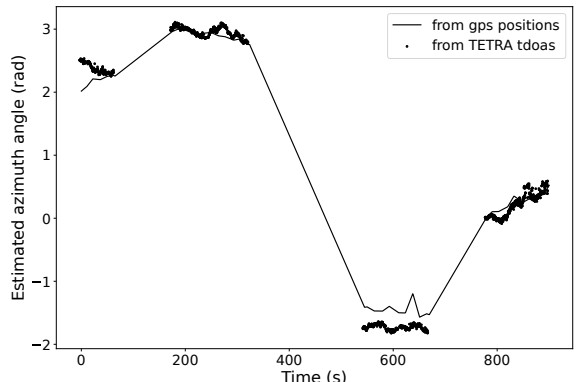
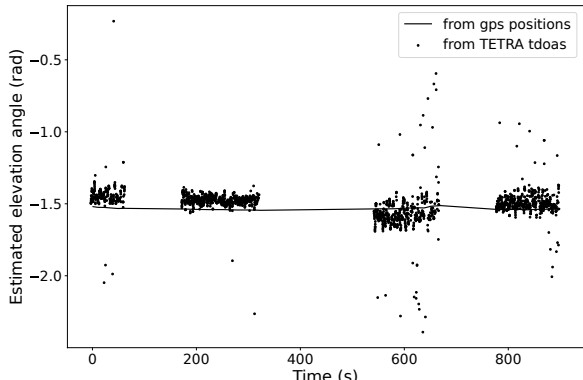

**Figure 8.** Angular comparison of azimuths (left) and elevation (right) between GPS references and TETRA estimates during a calibration experiment. *The emitter's elevation is interpolated from the immersion depth of the TETRA antenna.*

### 4.2 Whistle contours

Whistle contours were determined using DYOC (Lehnhoff et al., in review.a), a deep learning tool made for whistle contour annotation and developed using parts of this dataset. Then, contours were manually cleaned using PyAVA, a custom-made annotation tool in Python. An example of the results is shown in Fig. 9.

In total, 4,637 annotated whistle contours were verified and are provided as .json files (dictionaries) with the audio recordings. In each file, different keys correspond to different whistle contours, with points represented as lists of time-frequency coordinates. This dataset provides ground truths that could be used to study short-beaked common dolphin whistle repertoire, and to train and/or test performances of models for the extraction of whistle contours.

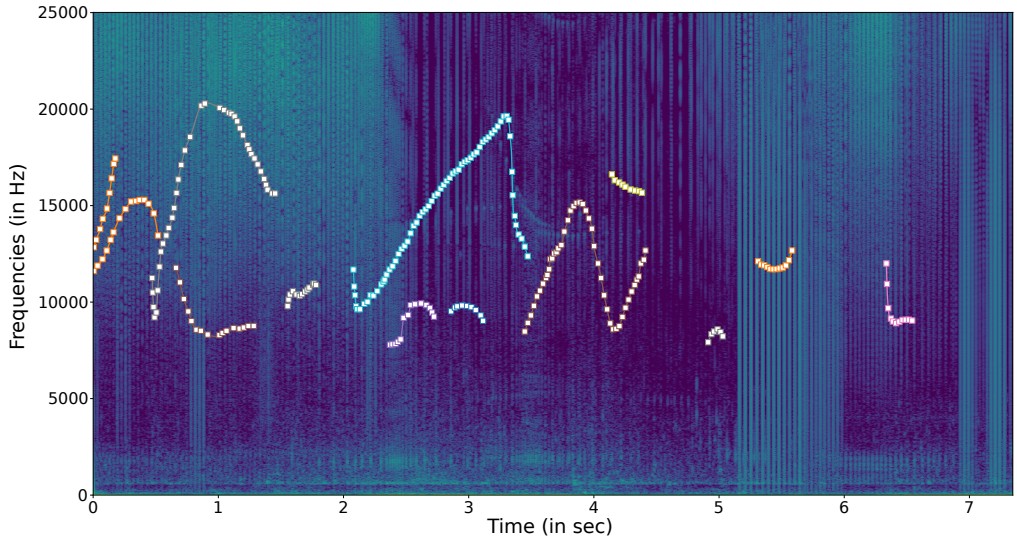

**Figure 9.** Screenshot showing whistle contour annotations of audio file "SCW1807_20200712_090400.wav".

## 5 Conclusions

Overall, this open-access dataset is the first to provide high-quality recordings of free-ranging short-beaked common dolphins (*Delphinus delphis*) in various observed behavioural states. The size of this dataset is substantial, with over 400 minutes of acoustic recordings of wild animals at sea, containing about 68,000 echolocation clicks, 4,600 whistles and 350 pulsed sounds. The combination of recordings from a single hydrophone and a tetrahedral array of hydrophones, along with detailed behavioural observations and manual whistle annotations, enables diverse research applications including acoustic repertoire analysis, behavioural studies and sound source localisation. Future work could leverage this data to better understand the relationship between dolphin vocalizations and specific behaviours, improve automated whistle detection techniques, and advance our knowledge of dolphin communication systems in general.

*Code and data availability.* Data described in this manuscript can be accessed at zenodo.org/records/14637675 under data DOI: 10.5281/zenodo.14637674 (Lehnhoff, 2025). Reuse of this dataset is facilitated by several scripts available at gitlab.lis-lab.fr/dolphinfree-experiments.





## Appendix A: Observation table

Visual observations were written down by observers on board using the following table document (Fig. A1).

**Date :**                    **Observation plateform :**                    **Observers :**

| TIME | GROUP | SEQUENCE | PINGER | | NET | GENERAL BEHAVIOUR | | | | | INT. | SPECIFIC BEHAVIOUR | | | | | DIST. | NUM. | SPACE | DIR. | SPEED | DIVE TIME | BOATS | NOTES |
|------|-------|----------|--------|--|-----|-------------------|--|--|--|--|------|--------------------|--|--|--|--|-------|------|-------|------|-------|-----------|-------|-------|
| HH:MM | # ID | # seq | pinger 1: pre- (off) 2: on 3 : post- | type | Orientation | Yes (1) No (0) | Travel. % | Pred. % | Soc. % | Milling % | Attrac.% | Response intensity to pinger 0, 1, 2 | Simult. surfacing % | Active surfacing % | Blows % | Dive % | Others (Jump, Lobtail, Spyhopping) % | Observers to dolphin group (m) | Number of dolphins in group | Group spacing: Compact. Scattered | Direction of the group: Variable Continuous | Slow (<10 kt) Fast (>10 kt) | Std. <2 min Variable Long >2 min | Number of boats present in a 1 NM radius | **Additional observations** (including sudden change in direction, speed and/or dive time, etc.). Start/End time of pinger transmission; hydrophone launch. |
| | | | | | | | | | | | | | | | | | | | | | | | | |
| | | | | | | | | | | | | | | | | | | | | | | | | |
| | | | | | | | | | | | | | | | | | | | | | | | | |
| | | | | | | | | | | | | | | | | | | | | | | | | |
| | | | | | | | | | | | | | | | | | | | | | | | | |
| | | | | | | | | | | | | | | | | | | | | | | | | |
| | | | | | | | | | | | | | | | | | | | | | | | | |
| | | | | | | | | | | | | | | | | | | | | | | | | |
| | | | | | | | | | | | | | | | | | | | | | | | | |
| | | | | | | | | | | | | | | | | | | | | | | | | |
| | | | | | | | | | | | | | | | | | | | | | | | | |
| | | | | | | | | | | | | | | | | | | | | | | | | |
| | | | | | | | | | | | | | | | | | | | | | | | | |
| | | | | | | | | | | | | | | | | | | | | | | | | |
| | | | | | | | | | | | | | | | | | | | | | | | | |
| | | | | | | | | | | | | | | | | | | | | | | | | |

**Figure A1.** Document used to record visual observations during surveys.



**Appendix B: Technical details**

**B1 icListen HF Hydrophone**

The icListen HF Hydrophone is produced by ©Ocean Sonics. Its specifications are available on their website (link). In the following figures, we report the hydrophone specifications provided by the manufacturer.

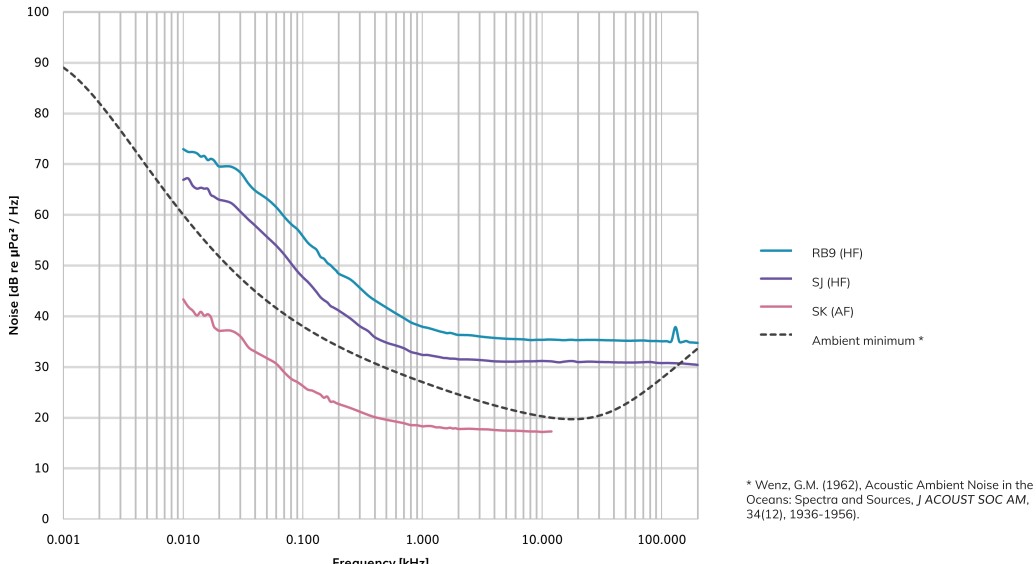

**Figure B1.** Noise spectrum levels of different iclisten HF hydrophones, provided by ©Ocean Sonics.

*The model SJ (purple) was the one used in this study.*

| | SJ | RB | SK | Units |
|---|---|---|---|---|
| **SIGNAL PERFORMANCE** | | | | |
| Low Frequency Cutoff | 10 | 10 | 10 | Hz |
| ±3 dB Bandwidth | 200 | 200 | 12.8 | kHz |
| Usable Bandwidth | 200 | 200 | 12.8 | kHz |
| Maximum Data Sample Rate | 512 | 512 | 32 | ksps |
| Minimum Data Sample Rate | 1 | 1 | 1 | ksps |
| Resolution | 24 | 24 | 24 | bits |
| Minimum Self Noise | 32 | 36 | 17 | dB re µPa²/Hz |
| Peak Input Level (µPa) | 180 | 184 | 168 | dB re µPa |
| Peak Input Level (Volts) | 6 | 6 | 6 | dBV |
| Voltage Sensitivity | -173 | -177 | -162 | dB re V/µPa |

**Figure B2.** Signal performance table of different iclisten HF hydrophones, provided by ©Ocean Sonics.

*The model SJ (purple) was the one used in this study.*




## B2 CR3 hydrophone

The CR3 hydrophone is produced by Cetacean Research™. Its specifications are available on their website (link). In the following figures and tables, we report the hydrophone specifications provided by the manufacturer.

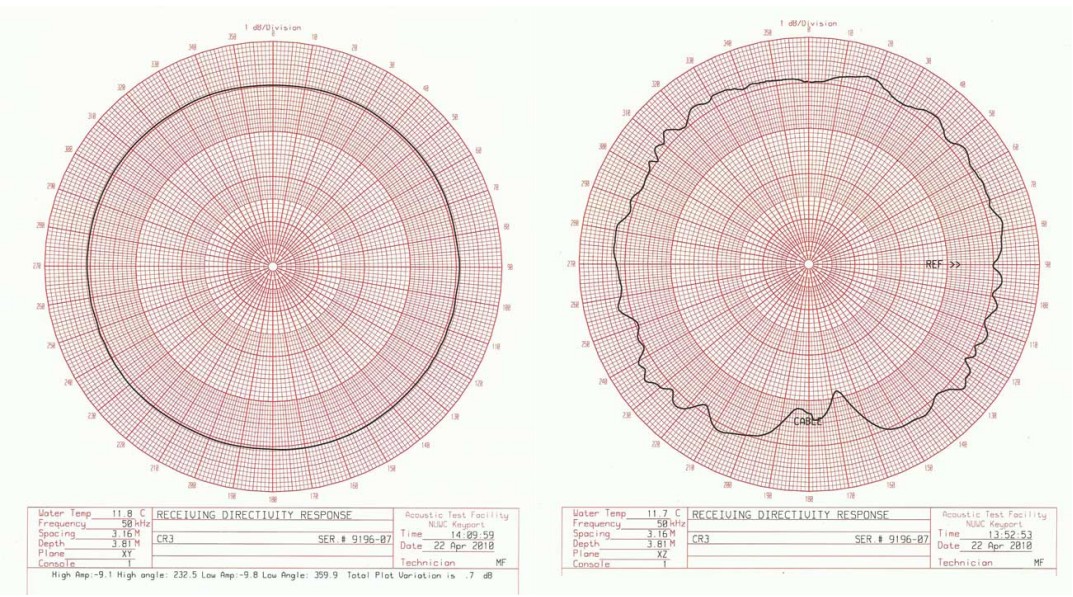

**Figure B3.** Horizontal (left) and vertical (right) beam pattern of the CR3 hydrophone, provided by Cetacean Research™.

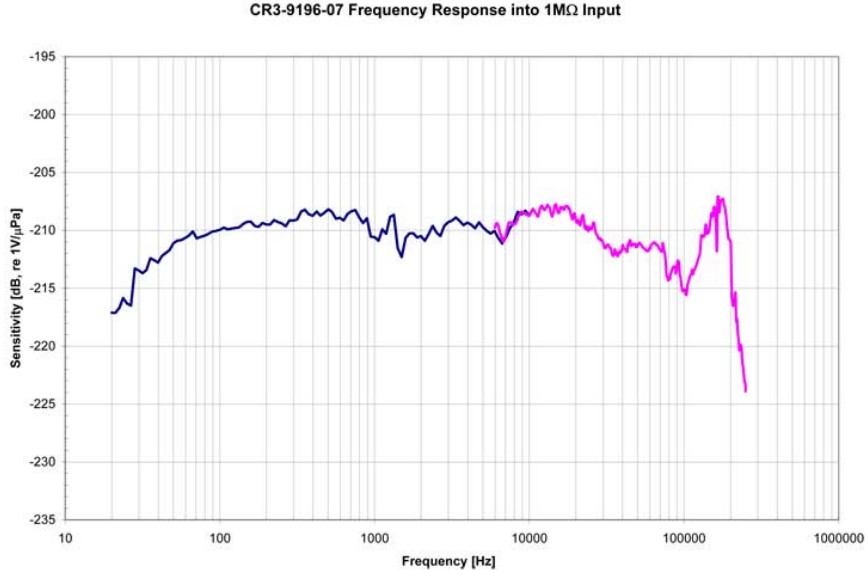

**Figure B4.** Frequency response of the CR3 hydrophone, provided by Cetacean Research™.

| Parameter | value |
|---|---|
| Linear Frequency Range ($\pm$3dB) | 0.0004 to 180 kHz |
| Usable Frequency Range (+3/-12dB) | 0.0001 to 240 kHz |
| Transducer Sensitivity | -207 dB re 1V/$\mu$Pa |
| SPL Equiv. Self Noise at 1kHz | 54 dB re 1$\mu$Pa/$\sqrt{}$Hz |
| Maximum Operating Depth | 980 m |
| Operating Temperature Range | -40 to 90 °C |
| Capacitance | 6.7 nF |
| Dimensions | 50 mm L x 18 mm dia |

**Table B1.** CR3 hydrophone specifications, provided by Cetacean Research™.

## B3 SQ26 Hydrophone

The SQ26 hydrophone is produced by Cetacean Research™. Its specifications are available on their website (link). In the following figures and tables, we report the hydrophone specifications provided by the manufacturer.

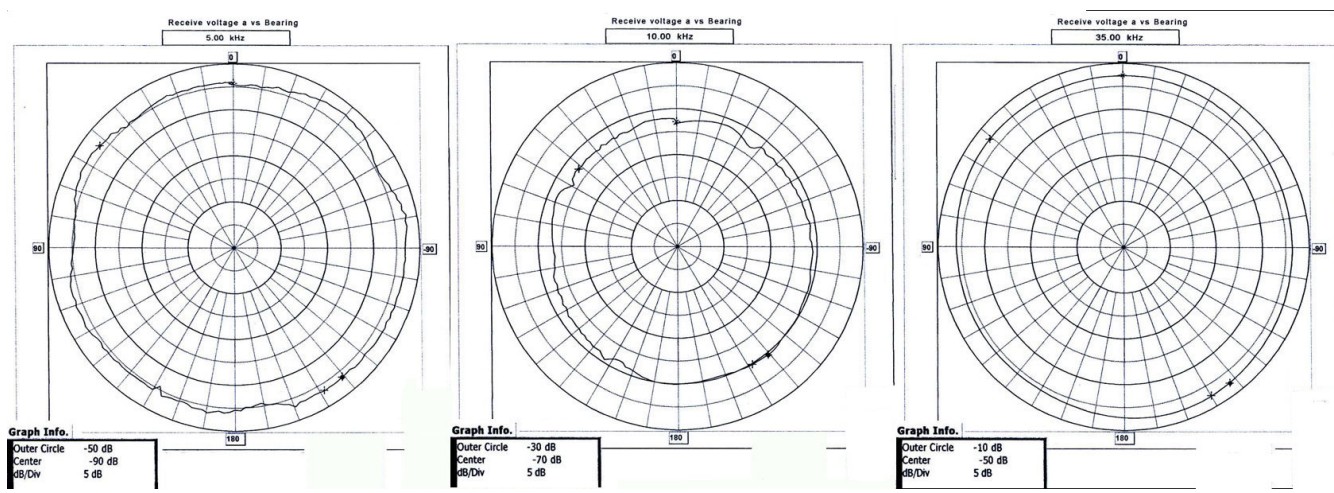

**Figure B5.** Horizontal beam pattern at 5 kHz (left), 10 kHz (centre), and 35 kHz (left) of the SQ26 hydrophone, provided by Cetacean Research™.



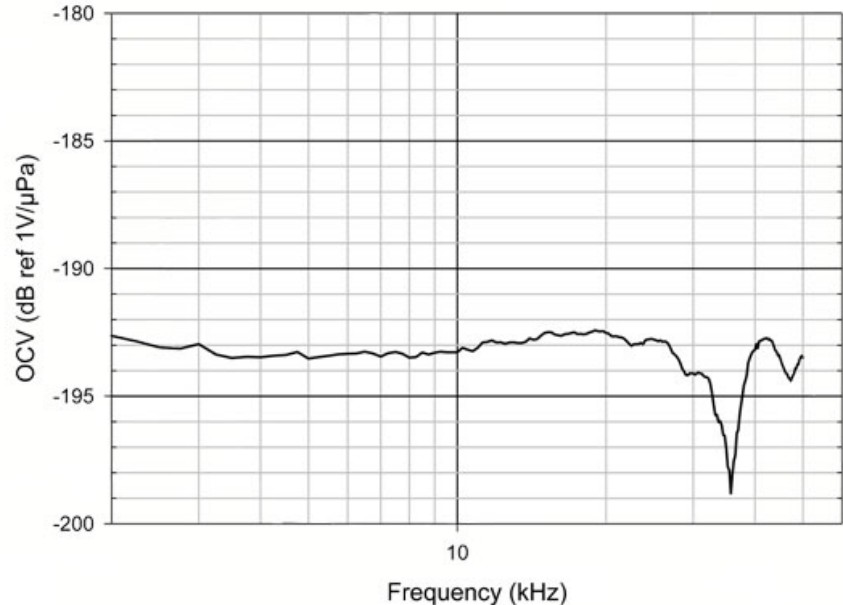

**Figure B6.** Frequency response of the SQ26 hydrophone, provided by Cetacean Research™.

| Parameter | value |
|---|---|
| Linear Frequency Range ($\pm$1dB) | 0.001 to 28 kHz |
| Usable Frequency Range (+3/-12dB) | 0.001 to > 50 kHz |
| Transducer Sensitivity | -193.5 dB re 1V/$\mu$Pa |
| SPL Equiv. Self Noise at 1kHz | 54 dB re 1$\mu$Pa/$\sqrt{}$Hz |
| Maximum Operating Depth | down to 2,000 m |
| Operating Temperature Range | -30 to 60 °C |
| Capacitance | 1.4 nF |
| Dimensions | 25.4 mm L x 25.4 mm dia |

**Table B2.** SQ26 hydrophone specifications, provided by Cetacean Research™.



*Author contributions.* Conceptualisation: B.M., H.G. and L.L.; methodology: B.M., H.G., L.L., Y.L.G., E.M. and O.V.C.; TETRA design: H.G.; Code, formal analysis: L.L., H.G. and B.M.; validation, supervision: B.M. and H.G.; data curation: B.M. and L.L; writing—original draft preparation: L.L.; figures: L.L. and O.V.C.; data acquisition, writing—review and editing: all authors; project administration: B.M.; project funding acquisition: B.M. and H.G. All authors reviewed the manuscript.

*Competing interests.* The authors declare no conflict of interest. The funders had no role in the design of the study; in the collection, analyses, or interpretation of data; in the writing of the manuscript, nor in the decision to publish the results. The decision to publish only a part of the collected data was prompted by the presence of signals protected by intellectual property.

*Disclaimer.* The DOLPHINFREE project has been approved by agreement 0-12520-2021/PREMAR_ATLANT/AEM/NP from the French Maritime Prefecture of the Atlantic "to conduct a survey for monitoring groups of common dolphins by means of scientific instruments off the south Finistère coast, following Décret n°2017-956 of the scientific marine research".

*Acknowledgements.* We thank Paul Best, Sophie Boyer and Eleonore Meheust for their assistance in collecting some of the data made available with this paper. We thank Michael Paul for improving the English of the paper.



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
