# Peer review of "High resolution acoustic recordings of wild free-ranging short-beaked common dolphins for etho-acoustical and repertoire studies"

_Earth System Science Data, 2025_

## Author Response (AR1)

Note: Original referee's comments are in black italic font. Author's responses are in blue.

18 June 2025

We have received the comments from 2 Referees on the interactive discussion page dedicated to the preprint of our manuscript. Overall, they found that the manuscript was *useful and of interest*, and *met the requirements for this type of journal.*
Referee #1 suggested *a revision of the English language, the removal of several repetitions, and some developments.* In particular, he/she/they found *the Introduction quite confused, failing to give a comprehensive picture* of the context surrounding *D. delphis sounds and correlation with behaviour.* In addition, he/she/they thought the *conclusion would benefit from a paragraph that reinforces the key findings, their broader significance, and potential applications.*
Referee #2 suggested corrections for f*ew typos and grammatical errors.* The *only concern was regarding behavioural data,* more specifically *regarding biases and interpretation.*

We thank both referees for their constructive comments, each of them is answered below.

**#1 Referee Comments**

*This manuscript describes open-access datasets of acoustic recordings (about 6.5 hours) of wild free-ranging short-beaked common dolphins (Delphinus delphis) collected in the northern Bay of Biscay, France, from summers of 2020 to 2022 during various behavioural states (including foraging, travelling, socializing, milling, and attraction to the boat).*
*The topic covered meets the objectives of the journal and is potentially useful and of interest to some of the journal's readers. However, this manuscript does not yet meet the quality standards required for acceptance for publication, as some developments are necessary, including revision of the English language and the removal of several repetitions.*

We appreciate the overall positive assessment of our work, and we thank the referee for the constructive and detailed comments. Below, we provide responses to each one of the comments. Please be assured that all corresponding changes have been incorporated in the revised version of the manuscript. In addition, to improve clarity and style, the manuscript has been reviewed by a native English speaker with experience in scientific writing.

*SPECIFIC COMMENTS*
*Abstract*
*Line 9-11: The sentence is a repetition of the previous one.*
Thank you for pointing this out. The repeated sentence has been removed to improve clarity.

*Introduction*
*In my opinion, in its current state, the Introduction is quite confused, fails to give a comprehensive picture (details on the acoustic characteristics of D. delphis sounds, i.e.*

*clicks, whistles and burst pulses, are missing, as well as correlation with behaviour) and needs English editing. Please consider the following suggestions.*

The Introduction has been reorganized for improved clarity and structure. A more extensive description of *D. delphis* clicks, whistles, and burst-pulsed sounds will be added, along with known correlations with behavior.

*Line 19-23: Please consider to remove the entire part or to use a portion of it at the end (see comments to line 64-65).*

This section has been moved to the end of the Introduction, as suggested [line 53-55].

*Line 23-26: Please start the Introduction with "This data descriptor .... D. delphis"*

The Introduction now begins with "This data descriptor…" [line 17].

*Line 26-27: Please consider to remove the sentence "The present study ... each recording" as it seems a repetition and it is not clear what you mean with "high-sapling".*

The phrase "high-sampling" referred to the high sampling rate of each recording. This sentence has been revised for clarity and to avoid redundancy [line 20].

*Line 29: Please consider to remove "including short-beaked common dolphin".*

The words "including short-beaked common dolphin" [line 21] have been removed.

*Line 35: Please check English ("that form an other form").*

This formulation was removed as lines 24-39 were modified.

*Line 35-36: Please consider to mitigate this sentence on communication function (at line 39 you correctly say that the function is still under investigation).*

The sentence has been restructured [line 33-39] to reflect that the communication function of pulsed sounds is still under investigation.

*Line 36: Please add here a ref (e.g. Herman & Tavolga, 1980) after "pulsed sounds".*

 "Pulsed sounds" are now better defined, and the suggested reference was added [line 33].

*Line 36-38: Please consider to re-phrase these sentences, possibly starting with ICI. Burst pulsed sounds were defined as "graded signals" by Murray et al. (1998), and that they were often recorded in arousal contexts. Then, before starting the description of the dataset (from line 40), please consider to add info/details on the acoustic characteristics of D. delphis sounds and correlation with the behaviour.*

This section now includes a more detailed description of burst-pulsed sounds, including their context and characteristics, as well as their known behavioral associations in *D. delphis* [lines 24-39].

*Line 43-44: Please check English.*

This sentence has been simplified for clarity [Line 42-45]

*Line 51-52: It seems a repetition.*
The repetition has been removed.

*Line 52-62: Please consider to remove this part as it seems too methodological for the Introduction. I also suggest to move figure 1 at the end of line 95.*
These lines, along with Figure 1 (now figure 3), have been moved to the "Materials and Surveys" section [line 69-74], as they were too specific for the Introduction section.

*Line 63-64: Again, it seems a repetition.*
The repeated sentence has been removed.

*Line 64-65: Please reshape the sentence to clearly underline the worth of the dataset, also contextualizing its usefulness in the context mentioned in line 19-23.*
The last paragraph of the Introduction [line 53-58] has been updated, using the context previously mentioned at the beginning of the introduction.

***Material and surveys***
*Line 69: Please consider to renumber the figures if you move figure 1.*
Figures have been renumbered to reflect the revised structure of the manuscript.

*Line 70-74: Please check English. Please add a ref to the study area in the figure legend.*
The sentences [line 61-67] have been rewritten for clarity and conciseness. The figure legend [Figure 1] now includes the name of the study area

*Line 79: Please consider to replace the word "parameters" with "procedures".*
The term "parameters" has been replaced with "procedures" [Line 79].

*Line 80: It is not clear to me what the Author mean with "computational processing". Please better define it.*
The term "without any computational processing" has been removed, as it was simpler to simply use the term "raw acoustic recordings" [Line 80].

*Line 90: The use of a fishing nets during your scientific surveys may raise ethical issues. Have you obtained an opinion from an ethics committee and an authorization to proceed from the government/local authorities? Is this document available? May we consider the Disclaimer at line 212? Please provide details on this relevant question.*
It seems that the disclaimer (lines 212–214) added in the initial version of the manuscript was missing its second part. The disclaimer has been updated. Specifically, we added the following sentence: *"The DOLPHINFREE project obtained a favorable opinion from the Ethical Committee for Animal Experimentation of Languedoc Roussillon (CEEA-LR) for request #26568."* However, these documents are not available online.

*In addition, please give specifics on the net characteristics (mesh size, length, etc) and on how you used it.*

Additional details about the fishing net (e.g., mesh size, length, and material) have been included in Appendix B1 [line 211-214], with a reference in text [Line 114].

**Conclusions**

*Conclusions would benefit from a paragraph that reinforces the key findings, their broader significance, and potential applications.*

The Conclusions [Line 194-205] now includes a more comprehensive paragraph reinforcing the value of the dataset, its potential applications in behavioral and conservation research, and its contribution to open-access marine mammal acoustic data.

**#2 Referee Comments**

*The paper describes the necessary information for what will likely prove a useful acoustic dataset from common dolphins. It seems to meet the requirements for this type of journal. The introduction provides useful background on the acoustic signals of common dolphins, and the relevant information to understand the dataset that is being made available. The methods sections are comprehensive and well-explained. The results provide useful examples of the nature of the data available; Figure 5 for example is excellent.*

We thank the referee for the constructive and overall positive comments on our manuscript. We particularly appreciate your recognition of the usefulness of the acoustic dataset, the clarity of the manuscript sections, and the quality of the figures. In the final version, the Introduction and Methods sections are restructured based on comments from all referees.

*My only concern is regarding the behavioural data. Assessing behavioural states is somewhat subjective, and potentially biased if not all animals in a group are performing the same behaviour. Furthermore, it is possible that individuals could be performing two types of behaviour at the same time, e.g. socialising while being attracted to the boat. I think adding a few caveats regarding interpretation of the behavioural data would be useful.*

To address concerns regarding behavioural data, we have added the following aspects in the revised version of the manuscript and in the dataset available through Zenodo:
(i) The data was initially collected as relative percentages of each behavioural state, to reflect the different behaviours exhibited within a group of dolphins and/or the several behaviours exhibited by a same individual (mainly attraction & socialisation during our surveys).
(ii) As a result, text in sub-section 'Observation notes' [lines 123-128] and Table 2 have been updated to precise the behavioural data collected.
(iii) The dataset available on Zenodo has been updated to include the initial relative percentage of each behavioural state recorded during the surveys

*There are few typos and grammatical errors which need to be corrected – see below.*
*Line 5: change to "observational data"?*
The typo was corrected [Line 5].

*Line 10: sentence is a repeat of a previous one.*
The repeated sentence was removed [Line 10].

*Line 16: "two dimensional movements" sounds a bit odd on this context. Perhaps clarify what you mean.*
We agree that "two-dimensional movement" may be unclear in this context and does not accurately reflect the intention of the sentence. Since the dataset includes audio recordings from four synchronised hydrophones, the directionality of sounds can be analysed using polar coordinates (elevation and azimuth angles), hence the two dimensions. However, to improve clarity, this sentence was revised to only mention "directionality" [Line 14].

*Line 35: change to "another".*
This sentence was removed during the restructuring of the introduction

*Line 40: change to " ... performed during the DOLPHINFREE project ...".*
Change added to the revised version [line 40].

*Line 49: change to " .. data that are not ...".*
Change added to the revised version [line 51].

*Line 68: change to "non-systematic".*
Change added to the revised version [line 61].

*Line 110: change to "bycatch occurs".*
Change added to the revised version [line 110].

*Line 120: change to "which in total add up to ".*
Change added to the revised version [line 133-134].

*Line 128: change to "observation data were collected".*
Change added to the revised version [line 119].

*Line 130: delete extraneous bracket.*
Bracket removed [line 121].

*Table 2" change "signification" to "description"?*
The title [Line 128] has been changed to: *"Description of column headers in visual observation data (XLSX) files".*

*Line 136: change to "audio recordings".*
Change added to the revised version [line 147].

Once again, we sincerely thank all referees for their helpful and thoughtful feedback.